# Evaluation of Breast Cancer Size Measurement by Computer-Aided Diagnosis (CAD) and a Radiologist on Breast MRI

**DOI:** 10.3390/jcm11051172

**Published:** 2022-02-22

**Authors:** Ji Yeon Park

**Affiliations:** Department of Radiology, Inje University Ilsan Paik Hospital, Goyang 10380, Korea; zzzz3@hanmail.net

**Keywords:** magnetic resonance imaging, breast neoplasm, computers, diagnosis, radiologists

## Abstract

Purpose: This study aimed to evaluate cancer size measurement by computer-aided diagnosis (CAD) and radiologist on breast magnetic resonance imaging (MRI) relative to histopathology and to determine clinicopathologic and MRI factors that may affect measurements. Methods: Preoperative MRI of 208 breast cancers taken between January 2017 and March 2021 were included. We evaluated correlation between CAD-generated size and pathologic size as well as that between radiologist-measured size and pathologic size. We classified size discrepancies into accurate and inaccurate groups. For both CAD and radiologist, clinicopathologic and imaging factors were compared between accurate and inaccurate groups. Results: The mean sizes as predicted by CAD, radiologist and pathology were 2.66 ± 1.68 cm, 2.54 ± 1.68 cm, and 2.30 ± 1.61 cm, with significant difference (*p* < 0.001). Correlation coefficients of cancer size measurement by radiologist and CAD in reference to pathology were 0.898 and 0.823. Radiologist’s measurement was more accurate than CAD, with statistical significance (*p* < 0.001). CAD-generated measurement was significantly more inaccurate for cancers of larger pathologic size (>2 cm), in the presence of an extensive intraductal component (EIC), with positive progesterone receptor (PR), and of non-mass enhancement (*p* = 0.045, 0.045, 0.03 and 0.002). Radiologist-measured size was significantly more inaccurate for cancers in presence of an in situ component, EIC, positive human epidermal growth factor receptor 2 (HER2), and non-mass enhancement (*p* = 0.017, 0.008, 0.003 and <0.001). Conclusion: Breast cancer size measurement showed a very strong correlation between CAD and pathology and radiologist and pathology. Radiologist-measured size was significantly more accurate than CAD size. Cancer size measurement by CAD and radiologist can both be inaccurate for cancers with EIC or non-mass enhancement.

## 1. Introduction

Breast conserving surgery has become the standard treatment strategy for early-stage, operable breast cancer, thus, the physician’s preoperative treatment planning is critical. The precision of the excision is an important factor in achieving local tumor control and patient survival [1]. Therefore, accurate tumor size measurement is essential for planning of breast conserving surgery and its successful completion as well as for prediction and evaluation of chemotherapy responsiveness [2,3]. 

Magnetic resonance imaging (MRI) is known to be more accurate than mammography or ultrasonography for assessment of breast cancer size [3,4,5]. MRI measurement accuracies have ranged between 43–88% [4,5,6,7,8]. However, evaluation of breast cancer by MRI takes significant time for image processing and interpretation. Inter-and intra-observer variability, furthermore, are additional drawbacks [9,10,11]. Various computer-aided diagnosis (CAD) systems for breast MRI have been devised and made commercially available to address these limitations and accelerate image processing and analysis. Assessment of CAD-generated tumor extent has been reported in various settings. One study reported a concordance rate of 44.1% for tumor size measurement by MRI CAD [12]. CAD measurement was revealed to be more accurate than measurement by radiologists in 57 breast cancer patients receiving neoadjuvant chemotherapy [13]. On the other hand, DeMartini et al. reported that CAD sizes were less accurate than those measured by radiologist in predicting the size of 16 residual breast malignancies after neoadjuvant chemotherapy [14]. Song et al. determined that CAD was feasible for assessment of tumor extent in 86 invasive breast cancers [15], and Levrini et al. found that MRI CAD size assessment of 52 breast lesions (including benign and malignant lesions) was as accurate as measurement performed on MRI images [16].

To our best knowledge, no study has examined breast cancer size measurement by CAD and a radiologist relative to histopathology in a large number of breast cancer cases. Therefore, the aim of this study was to assess and compare the correlation of breast cancer size measurement between CAD and histopathology, and a radiologist and histopathology on breast MRI and to determine the clinicopathologic and MRI factors which may affect breast cancer size measurement by CAD or a radiologist.

## 2. Materials and Methods

### 2.1. Study Design and Patients

The institutional review board approved this retrospective study, and waived the need for written informed consent. 

A total of 1188 breast MRI examinations were performed at our institution between January 2017 and March 2021. We excluded MRI for evaluation of implants, postoperative follow-up and cases of preoperative neoadjuvant chemotherapy and cases that did not undergo surgery at our institution. Finally, preoperative MRI of 208 breast cancers in 202 patients (female:male = 200:2) were included in this study. Six patients had bilateral breast cancers.

Breast cancer was confirmed prior to the preoperative MRI examination by 14-gauge ultrasound-guided core biopsy or 9-gauge stereotactic biopsy. All of the patients in this study underwent breast surgery within a month from the preoperative MRI.

### 2.2. Breast MRI Technique

Dynamic contrast-enhanced breast MRI examinations were performed with a 3.0-T or 1.5-T system (SIGNA Architect, GE Healthcare, Milwaukee, USA or Avanto, Siemens, Germany) and a dedicated 16-channel or 4-channel SENSE breast coil, with the patient in the prone position. Firstly, unenhanced T2-weighted propeller short T1 inversion recovery (STIR) or turbo inversion recovery magnitude (TIRM) axial images and T1-weighted spin echo axial images were obtained, and then 0.1 mmol/kg gadoteridol (15 mL ProHance; Bracco imaging, Singen, Germany) was administrated via intravenous injection. Dynamic contrast-enhanced examinations included one pre-contrast and four post-contrast bilateral axial image acquisitions using a fat-suppressed T1-weighted 3D fast field echo sequence (TR/TE;7.0/1.7 300 × 280 matrixes, field of view; 360 × 360 mm, slice thickness; 2 mm, no gap for 3.0-T, TR/TE; 4.8/2.4, 384 × 338 matrixes, field of view; 380 × 380 mm, slice thickness; 1 mm, no gap for 1.5-T). Four post-contrast image series were obtained at 90, 180, 270, 360 s after contrast administration. In all studies, dynamic subtraction (i.e., post-contrast images minus precontrast images) and 3D maximum intensity projection images were generated. The approximate total time per examination was 30 min.

### 2.3. CAD Analysis

All of the MRI examinations were subsequently processed by a commercial CAD system (CADstream version 6.0 Merge Healthcare, Chicago, IL, USA). The CAD system compared the pixel signal intensity values on the pre-contrast, first post-contrast and last post-contrast series. When the areas of enhancement met a 50% enhancement threshold for initial enhancement, they were automatically identified by color overlays on all MRI slices. For kinetic analysis, the initial phase patterns were determined by the signal change between the pre-contrast and initial post-contrast series and there were three patterns: slow (<50% increase), medium (50–100% at 90 s), and rapid (more than 100% at 90 s) enhancement. Delayed phase enhancement types were also categorized as persistent, plateau or washout. The persistent type, displayed in blue, represented pixel signal intensity with at least a 10% increase; the washout type, displayed in red, represented pixel signal intensity with at least 10% decrease; the plateau type, displayed in yellow, indicated pixel signal intensity with a less than 10% increase or a less than 10% decrease in the last post-contrast series relative to the first post-contrast series.

### 2.4. Size Evaluation of Breast Cancer

One radiologist with 10 years of experience in breast imaging measured the maximum diameter of the largest index cancer on axial and sagittal post-contrast and subtraction images of MRI. The same radiologist reviewed CAD data, and when the radiologist selected the largest tumor on the angio-map, the CAD system automatically generated the lesion size. When no lesion was present on MRI or MRI CAD, the size was set to 0 cm.

After analyzing the MRI, the pathological size was reviewed. The pathologic tumor size was defined as the maximum diameter of the area covered by the tumor, including surrounding in situ components, if present. The diameter of isolated carcinoma in situ was not included. If there were multifocal cancers on one side, we selected the largest diameter of the largest index tumor in the pathologic specimen.

The maximum lesion diameter by histopathology was used as the reference standard. The accuracy of the assessment of the tumor size by CAD or radiologist was calculated, and classified accordingly into accurate or inaccurate groups. Accurate estimation was defined as a 5 mm or smaller difference between CAD- or radiologist-determined size and the histopathologic size. The inaccurate group, meanwhile, was divided into over- or underestimation subgroups based on larger difference than 5 mm, respectively.

### 2.5. Histopathologic and Imaging Data Collection

The lesion type (mass or non-mass enhancement (NME)) and background parenchymal enhancement (BPE) on MRI were recorded, according to the Breast Imaging Reporting and Data System Atlas for MRI [17]. 

We reviewed the presence of associated carcinoma in situ component, extensive intraductal component (EIC), histologic grade, estrogen receptor (ER) status, progesterone receptor (PR) status, human epidermal growth factor receptor 2 (HER2), Ki-67 index, and molecular subtype (luminal A, luminal B, HER2(+), triple-negative). The histologic grade was graded as low or high. For Ki-67 labelling index, we set the boundary between high and low at 14% according to the St Gallen consensus [18].

### 2.6. Statistical Analysis

Pearson’s correlation coefficients were calculated to determine the association between the CAD or radiologist and the histopathologic size. The correlation coefficients were calculated, according to age, the presence of associated carcinoma in situ component, EIC, histologic grade, ER status, PR status, Ki-67 index, molecular subtype, morphologic type and BPE on MRI. The mean sizes were compared between the radiologist or CAD and histopathology using the paired t-test. In addition, agreement between the radiologist’s or CAD measurement and the histopathological size was assessed by the Bland-Altman plotting method. The mean difference between the two measurements and the 95% limits of agreement was calculated. The differences in clinicopathologic variables between accurate and inaccurate groups were evaluated using the chi-square test, Fisher’s exact test and Student’s t-test. Statistical analyses were performed using SPSS 25.0 (IBM, Armonk, NY, USA) and *p* < 0.05 was considered significant. 

## 3. Results

A total of 208 breast cancers of 200 women and two men (mean age: 57.62 years, range: 33–88) were included in this study and six patients had bilateral breast cancers. 154 lesions had undergone breast conserving surgery and 54 lesions had undergone mastectomy.

There were 137 (65.9%) invasive ductal carcinomas (IDC), 45 (21.6%) ductal carcinoma in situ (DCIS), nine (4.3%) mucinous carcinomas, seven (3.4%) invasive lobular carcinomas (ILC), three (1.4%) microinvasive carcinomas, three (1.4%) medullary carcinomas, two (1.0%) encapsulated papillary carcinoma, one (0.5%) mixed invasive ductal/lobular carcinoma and one (0.5%) invasive cribriform carcinoma.

### 3.1. Correlation between CAD-Generated Tumor Size or Radiologist-Measured Size and Histopathologic Size

The overall mean size as predicted by CAD was 2.66 ± 1.68 cm (range: 0–10.1 cm), the mean size as measured by the radiologist was 2.54 ± 1.68 cm (range: 0–9.8 cm), and the mean pathologic size was 2.30 ± 1.61 cm (range: 0.1–11.0 cm). Tumor measurements as determined by MRI CAD or the radiologist was significantly different from the pathological result (*p* < 0.001). Tumor size between the radiologist and MR CAD was significantly different (*p* = 0.012).

Table 1 shows the correlation coefficients of the size measurement by MRI CAD, the radiologists and pathology, according to the clinicopathologic factors. The overall correlation between radiologist and pathologic measurement was 0.898 (*p* < 0.001) (Figure 1A). The correlation between MR CAD and pathologic measurement was 0.823 (*p* < 0.001) (Figure 1B). The correlation between the radiologist and MRI CAD measurement was 0.925 (*p* < 0.001). For MRI CAD and the radiologist, the size correlation coefficients of older patients (≥50 years), larger pathologic size (>2 cm), no in situ component, no EIC, high grade, negative ER, negative PR, positive HER2, high Ki-67, non-mass enhancement type and moderate to marked BPE type were higher than those of the younger patients, smaller pathologic size, presence of in situ component, EIC, low grade, positive ER, positive PR, negative HER2, low Ki-67, mass type and minimal to mild BPE type.

The Bland-Altman plot showed that the mean measurement difference between the radiologist and pathologic size was 0.244 cm, with a 95% confidence limit of agreement of −1.215 to 1.704 cm (Figure 2A), the mean difference between pathologic tumor size and the MR CAD was 0.36 cm, with a 95% confidence limit of agreement of −1.565 to 2.284 cm and mean difference between (Figure 2B). The mean difference between the radiologist and MR CAD was 0.115 cm with a 95% confidence limit of agreement of −1.16 to 1.39 cm (Figure 2C).

### 3.2. Correlation of Accurate and Inaccurate Groups for Size Measurement by MR CAD and Radiologist

Table 2 compares the accurate and inaccurate groups for size measurement by MR CAD and the radiologist. Tumor size by MR CAD was accurate in 114/228 cancers (54.8%) and size by radiologist in 152 cancers (73.1%) (Figure 3). MR CAD overestimated the size of 77 cancers (37.0%) and underestimated that of 17 cancers (8.2%). The radiologist overestimated the size of 45 cancers (21.6%) and underestimated that of 11 cancers (5.3%) (Figure 4). The radiologist measured size more accurately than did MR CAD, and with statistical significance (*p* < 0.001).

Table 3 shows the results of the analysis of the accurate and inaccurate groups for size measurement by MR CAD. The mean CAD size and pathologic size of the accurate group were 2.02 cm, 1.97 cm, while those of the inaccurate group were 3.43 cm, 2.70 cm, and mean CAD size and pathologic size in inaccurate group were significantly larger than those of the accurate group (*p* < 0.001 and *p* = 0.002). MR CAD-generated size was significantly more accurate for the small pathologic size group (2 cm or smaller size) than for the larger pathologic size group (*p* = 0.045). MR CAD size was more accurate for cancers with no EIC, positive PR, and mass type than those with EIC, negative PR and NME type (*p* = 0.045, 0.03 and 0.002). However, cancer size assessment by MR CAD was not significantly associated with age, histologic grade, ER, HER2, molecular subtype, Ki-67 and BPE (*p* > 0.05).

Table 4 shows the results of the analysis of the accurate and inaccurate groups for size measurement by radiologist on MR. The mean radiologic size and pathologic size of the accurate group were 2.20 cm, 2.11 cm, while those of the inaccurate group were 3.48 cm, 2.80 cm; as evidenced, the mean radiologic size and pathologic size in the inaccurate group were significantly larger than those of the accurate group (*p* < 0.001 and *p* = 0.022). Radiologist-measured size was not significantly different between the large and small pathologic size groups (*p* = 0.065). Radiologist-measured size was more accurate for cancers with no in situ or EIC, negative HER2, and mass type than those with in situ or EIC, positive HER2 and NME type (*p* = 0.02, 0.008, 0.003 and <0.001). However, cancer size assessment by radiologist was not significantly associated with age, histologic grade, ER, PR, molecular subtype, Ki-67 and BPE (*p* > 0.05).

## 4. Discussion

Higher accuracy with MRI for lesion extent assessment than with either ultrasound or mammography has been reported [3,4,5]. However, inter- or intra-observer variations may affect precision of measurement. Recently, the accuracy of MRI evaluation has been improved by combination with, for example, 3D volume MRI data and, in fact, the CAD system has been reported as accurate as the measurement performed on MR images [13,16,19,20,21].

In the present study, we investigated the breast cancer size measurement by CAD and radiologist on MRI relative to histopathology and compared the accuracy rates between the two groups. The correlation coefficients of cancer size measurement by radiologist and CAD on MRI, with reference to corresponding pathological results, were 0.898 and 0.823, which showed a very strong correlation. The radiologist measured cancer size more accurately than did MR CAD, with statistical significance (73.1% vs. 54.8%, *p* < 0.001). Song et al. reported that manual measurement of MRI was a better modality for assessment of tumor extent than MRI with CAD (0.766 vs. 0.513) [15]. Similarly, a preliminary study by DeMartini et al. with 15 patients demonstrated that radiologist-measured sizes were more accurate in predicting the size of residual malignancy than those measured by CAD [14]. Although CAD-generated size is less accurate than that measured by radiologist, CAD has a number of advantages in detecting breast cancers. The CAD system enables the radiologists to handle large images with speed and provides them with an angio-map and colored imaging that can show correlations with kinetic curves of breast lesions [9].

In the present study, the mean difference between the pathologic tumor size and that generated by CAD was 0.36 cm and the mean difference between the radiologist-measured and pathologic size was 0.244 cm. These results are consistent with the tendency of MR CAD-and radiologist-measured size to overestimation, relative to pathology. Previously reported mean differences between MRI and pathology were 0.2–0.368 cm and MRI overestimated tumor size compared with mammography and ultrasound [22,23].

Size discordances of cancer with in situ component or EIC by breast MRI have been reported in many articles [4,5,7,24,25,26,27]. As EIC is a strong risk factor for postoperative recurrence, it is important to accurately identify the extent of cancer preoperatively [28,29]. Cancer with EIC was reported to be underestimated on MRI in 30% and overestimated in 22% [24]. In our study, inaccurate rates by MR CAD and radiologist for cancers with EIC were 60% (21/25) and 40% (14/35), respectively. Allen et al. reported that 65.2% of DCIS had inaccurate size on preoperative MRI and 71.4% of inaccurate lesions had been overestimated [27]. Previously reported inaccurate rates of size measurement for DCIS using MRI ranged from 28–50% [4,7,30] and, in our study, inaccuracy rates by MR CAD and radiologist were 47.7% (71/149) and 31.5% (47/149). Overestimation of tumor extent on MRI may occur due to enhancement of benign proliferative processes such as fibrocystic changes or adenosis adjacent to the cancer, while underestimation may occur in cases exhibiting weak tumor angiogenesis in the stroma around ducts involving DCIS or EIC [24,26,31]. In some cases, CAD can also include additional enhancing tissue adjacent to the tumor in the size assessment, thereby incurring overestimation. This is a function of the CAD processing algorithm, which identifies and measures “connected” pixels with significant enhancement. Underestimation of CAD size also has been found and could potentially be improved by specifying a lower enhancement threshold that might reveal larger areas of significant enhancement [14]. Our study also showed that MR CAD-generated size was significantly more inaccurate for cancers with in situ component and EIC.

The MR feature most frequently seen in EIC-positive tumors was a mass surrounded by segmental, ductal or linear enhancement [25,29]. In our study, discordance of cancers with non-mass enhancement type more frequently occurred than did mass type when generated by CAD or radiologist (66.7% and 54.8%), and 88.9% and 71.4% of cancers with non-mass enhancement had been reported to influence size-measurement discordance on MRI [6,32].

In our study, MR CAD-generated cancer size was significantly inaccurate for larger (>2 cm, more than T2 stage) pathologic sizes, and 38.1% (37/97) of the cancer larger than 2 cm was overestimated on CAD, while 14.4% were underestimated. This result is similar to previously published data indicating that MRI had a tendency to overestimate, particularly for patients with pathologic tumor size greater than 2 cm [33]. However, another study reported that CAD-generated size showed a higher concordance rate for patients with T3 (>5 cm) lesions [12]. The issue of accuracy of CAD-generated size relative to actual pathologic size will require further research.

Breast cancers with different ER, PR, and HER2 statuses have distinct prognoses and show variable responses to endocrine therapy, radiation therapy, and chemotherapy [34,35,36]. HER2 overexpression is known to be associated with increased angiogenesis and higher degrees of angiogenesis are less accurately estimated on breast MRI after neoadjuvant treatment [2,37]. HER2 negativity has been reported to be highly associated with the accuracy of breast cancer measurement [2,6,32]. These findings might explain why the inaccuracy of cancer size measurement by radiologist was associated with HER2 positivity in the present study.

BPE on breast MRI can obscure or mimic lesion enhancement and decrease the accuracy of breast MRI [38,39]. In an investigation of the impact of BPE on tumor size estimation, moderate and marked BPE were correlated with inaccurate estimation based on MRI [32]. However, our study did not show any statistical relation between BPE and size estimation by CAD or radiologist. This discordance may have resulted from the difference in the proportion of moderate/marked BPE between the two studies and our study’s smaller number of moderate/marked BPE (11% vs. 23.9%). Therefore, further studies including a large number of cases will be required for a more comprehensive evaluation of the effect of BPE on tumor size measurement by MR CAD or radiologist.

Our study has some limitations. First, this was a retrospective study, and as such, bias could have been incurred. Second, only one radiologist measured the cancer size on breast MRI. However, the radiologist had 10 years of experience in breast imaging and this can be an advantage, avoiding interobserver variability. Third, this study compared only CAD- and radiologist-measured size. In practice, combined measurement such as CAD as a first reading followed by radiologist can be helpful, so further study will be required in the future. Fourth, we did not consider the effect of interobserver variability among pathologists and the possible difference of 1.5T or 3T MRI for tumor size evaluation. Fifth, we did not include multifocal or multicentric cancers, and considered the size of only the index largest cancer.

## 5. Conclusions

In conclusion, comparison of breast cancer size measurement between CAD and pathology, and between a radiologist and pathology, showed very strong correlations. Radiologist-measured tumor size was more accurate than CAD-generated size. Cancer size measured by radiologist and CAD on MRI can be inaccurate for cancers with EIC and of the non-mass enhancement type.

## Figures and Tables

**Figure 1 jcm-11-01172-f001:**
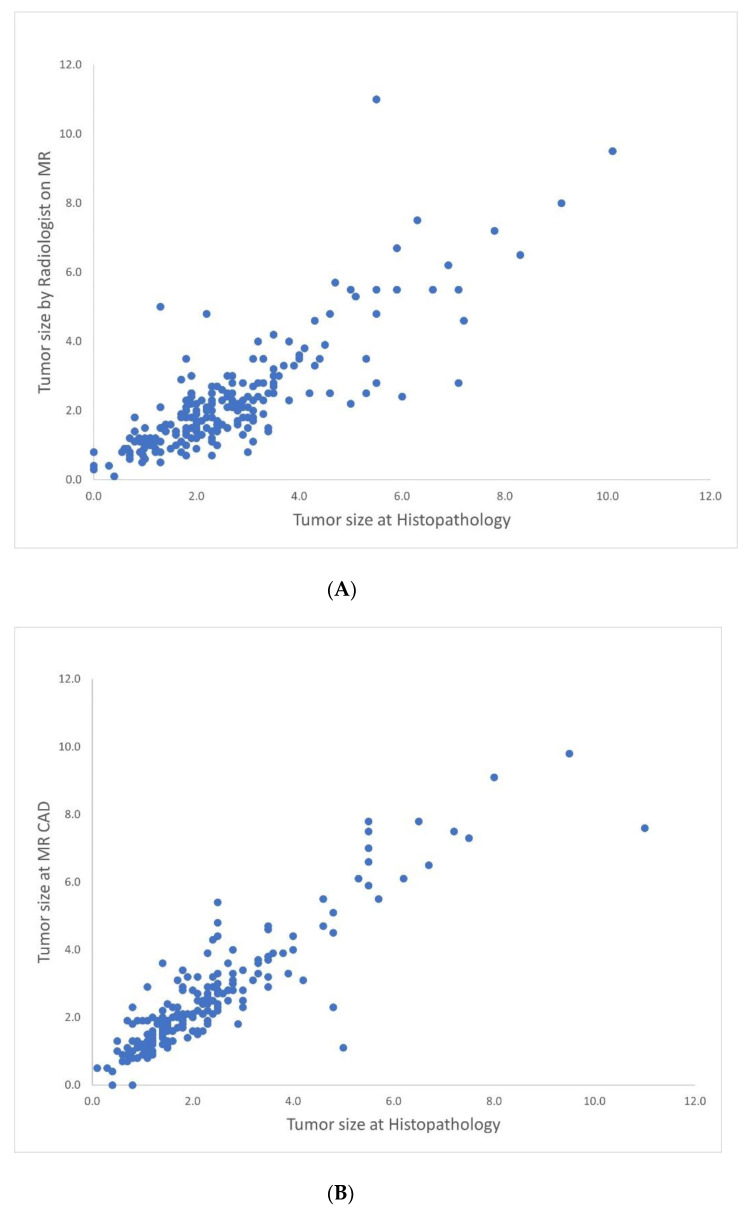
Scatter plot showing correlation between tumor size measurement obtained from CAD and radiologist, in reference to pathology. (**A**) Correlation between radiologist and pathology (r = 0.898) (**B**) Correlation between CAD and pathology (r = 0.823).

**Figure 2 jcm-11-01172-f002:**
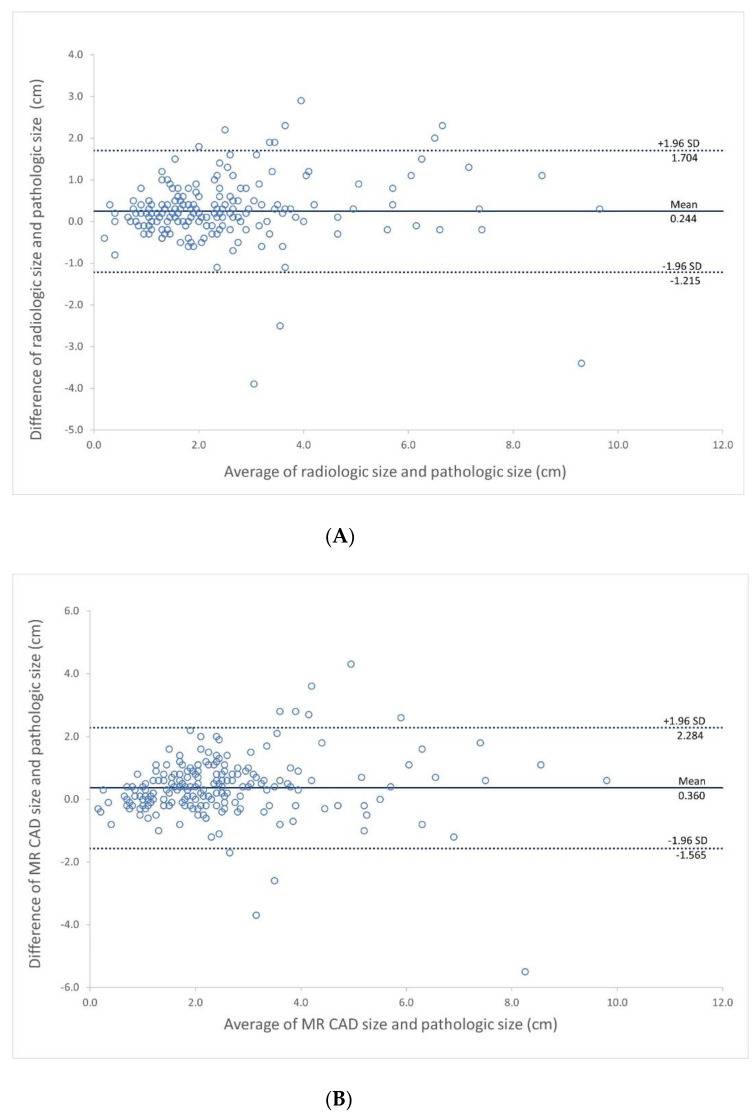
Bland-Altman plot showing agreement between tumor size measurements obtained from CAD and radiologist, with reference to pathology. (**A**) Agreement between radiologist and pathology (**B**) Agreement between MR CAD and pathology (**C**) Agreement between MR CAD and radiologist.

**Figure 3 jcm-11-01172-f003:**
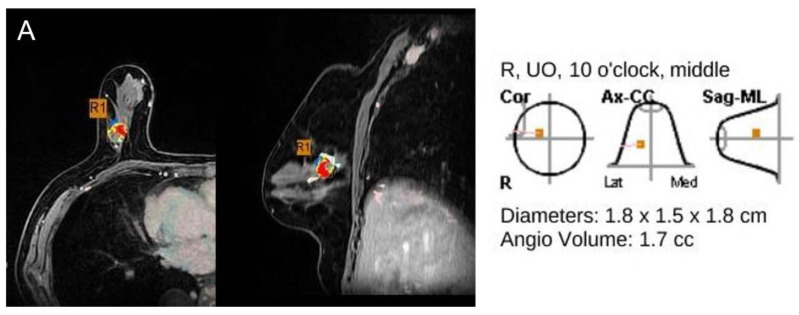
78-year-old woman with invasive ductal carcinoma in right breast. (**A**,**B**) Maximal diameter by CAD, radiologist and pathology were 1.8 cm, 2.1 cm, and 1.8 cm. Both CAD-and radiologist-measured sizes were accurate, in reference to pathology. This cancer was of luminal B subtype, negative EIC at pathology and mass type on MRI (R: right, UO: upper outer).

**Figure 4 jcm-11-01172-f004:**
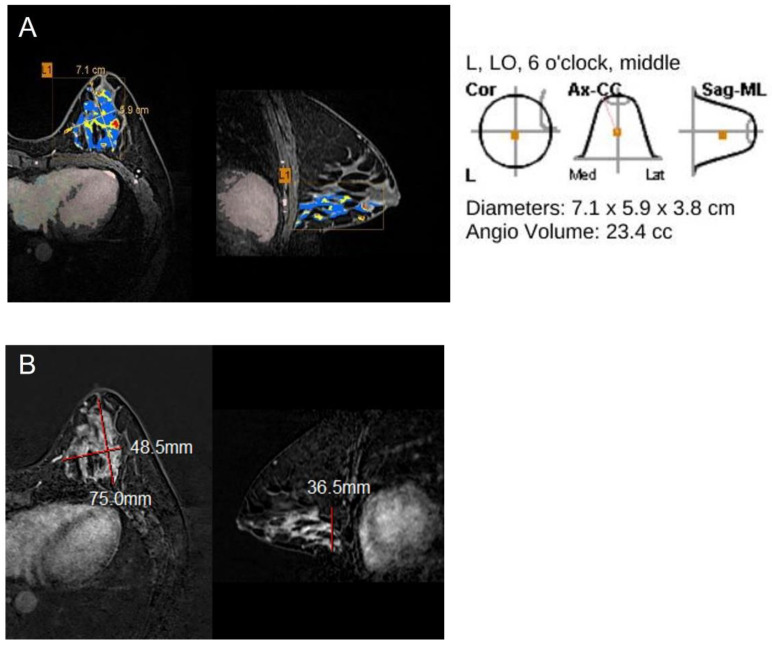
42-year-old woman with ductal carcinoma in situ in left breast. (**A**,**B**) Maximal diameter by CAD, radiologist and pathology were 7.1 cm, 7.5 cm, and 5.5 cm. Both CAD-and radiologist-measured sizes were inaccurate, in reference to pathology. This cancer was of HER2+ subtype, positive EIC at pathology and non-mass enhancement type on MRI (L: left, LO: lower outer).

**Table 1 jcm-11-01172-t001:** The correlation coefficients of size measurement by MR CAD, radiologist and pathology, according to the clinicopathologic variables.

Variables	N = 208	CAD-Pathology r (*p*-Value)	Radiologist-Pathology r (*p*-Value)	CAD-Radiologist r (*p*-Value)
Overall		0.823 (<0.001)	0.898 (<0.001)	0.925 (<0.001)
Age group				
<50 years	62	0.775 (<0.001)	0.893 (<0.001)	0.925 (<0.001)
≥50 years	146	0.850 (<0.001)	0.902 (<0.001)	0.925 (<0.001)
Pathologic size				
0–2 cm	111	0.605 (<0.001)	0.649 (<0.001)	0.812 (<0.001)
>2 cm	97	0.733 (<0.001)	0.854 (<0.001)	0.896 (<0.001)
In situ component				
(−)	59	0.923 (<0.001)	0.956 (<0.001)	0.953 (<0.001)
(+)	148	0.794 (<0.001)	0.883 (<0.001)	0.916 (<0.001)
NA	1			
EIC				
(−)	134	0.807 (<0.001)	0.898 (<0.001)	0.905 (<0.001)
(+)	35	0.783 (<0.001)	0.849(<0.001)	0.933 (<0.001)
NA	39			
Histologic grade				
Low	159	0.802 (<0.001)	0.885 (<0.001)	0.936 (<0.001)
High	47	0.888 (<0.001)	0.938 (<0.001)	0.898 (<0.001)
NA	2			
ER				
(−)	52	0.928 (<0.001)	0.950 (<0.001)	0.935 (<0.001)
(+)	156	0.779 (<0.001)	0.880 (<0.001)	0.921 (<0.001)
PR				
(−)	81	0.908 (<0.001)	0.940 (<0.001)	0.920 (<0.001)
(+)	127	0.776 (<0.001)	0.878 (<0.001)	0.929 (<0.001)
HER2				
(−)	160	0.778 (<0.001)	0.875 (<0.001)	0.920 (<0.001)
(+)	48	0.897 (<0.001)	0.941 (<0.001)	0.927 (<0.001)
Molecular subtype				
Luminal A	83	0.733 (<0.001)	0.865 (<0.001)	0.916 (<0.001)
Luminal B	74	0.825 (<0.001)	0.896 (<0.001)	0.921 (<0.001)
HER2	25	0.926 (<0.001)	0.941 (<0.001)	0.921 (<0.001)
Triple-negative	26	0.928 (<0.001)	0.960 (<0.001)	0.955 (<0.001)
Ki-67				
Low	93	0.744 (<0.001)	0.863 (<0.001)	0.921 (<0.001)
High	115	0.878 (<0.001)	0.923 (<0.001)	0.925 (<0.001)
MR finding				
Mass	164	0.782 (<0.001)	0.864 (<0.001)	0.903 (<0.001)
NME	42	0.786 (<0.001)	0.882 (<0.001)	0.908 (<0.001)
NA	2			
MR BPE				
Minimal	124	0.798 (<0.001)	0.857 (<0.001)	0.920 (<0.001)
Mild	61	0.836 (<0.001)	0.932 (<0.001)	0.934 (<0.001)
Moderate	13	0.890 (<0.001)	0.950 (<0.001)	0.927 (<0.001)
Marked	10	0.844 (0.002)	0.912 (<0.001)	0.966 (<0.001)
MR BPE				
Minimal + Mild	185	0.818 (<0.001)	0.894 (<0.001)	0.927 (<0.001)
Moderate + Marked	23	0.879 (<0.001)	0.934 (<0.001)	0.937 (<0.001)

NA: not available.

**Table 2 jcm-11-01172-t002:** Comparison of accurate and inaccurate group for size measurement by MR CAD and radiologist.

	CAD-Pathology	Radiologist-Pathology	*p*-Value *
Accurate	114 (54.8%)	152 (73.1%)	<0.001
Inaccurate	94 (45.2%)	56 (26.9%)	
Underestimation	17 (8.2%)	11 (5.3%)	
Overestimation	77 (37.0%)	45 (21.6%)	

*****: accurate vs. inaccurate.

**Table 3 jcm-11-01172-t003:** Analysis of accurate and inaccurate groups for size measurement by MR CAD.

Variables	Accurate 114 (54.8%)	Inaccurate 94 (45.2%)		*p*-Value *
		Underestimation 17 (8.2%)	Overestimation 77 (37.0%)	
Mean age (years)	58.38 ± 11.53	54.12 ± 15.81	57.26 ± 12.54	0.326
Age group				0.129
<50 years	29 (25.4%)	8 (47.1%)	25 (32.5%)	
≥50 years	85 (74.6%)	9 (52.9%)	52 (67.5%)	
Mean MR CAD size (cm)	2.02 ± 1.14	2.63 ± 1.91	3.61 ± 1.86	<0.001
Mean pathologic size (cm)	1.97 ± 1.12	4.08 ± 2.57	2.40 ± 1.72	0.002
Pathologic size				0.045
0–2 cm	68 (59.6%)	3 (17.6%)	40 (51.9%)	
>2 cm	46 (40.4%)	14 (82.4%)	37 (48.1%)	
In situ component				0.258
(−)	36 (31.6%)	3 (17.6%)	20 (26.0%)	
(+)	78 (68.4%)	14 (82.4%)	57 (74.0%)	
EIC				0.045
(−)	79 (69.3%)	8 (47.1%)	47 (61.0%)	
(+)	14 (12.3%)	6 (36.3%)	15 (19.5%)	
NA	21 (18.4%)	3 (17.6%)	15 (19.5%)	
Histologic grade				0.183
Low	84 (73.7%)	3 (17.6%)	61 (79.2%)	
High	30 (26.3%)	14 (82.4%)	14 (18.2%)	
NA			2 (2.6%)	
ER				0.148
(−)	33 (28.9%)	2 (11.8%)	17 (22.1%)	
(+)	81 (71.1%)	15 (88.2%)	60 (77.9%)	
PR				0.030
(−)	52 (45.6%)	2 (11.8%)	27 (35.1%)	
(+)	62 (54.4%)	15 (88.2%)	50 (64.9%)	
HER2				0.274
(−)	91 (79.8%)	12 (70.6%)	57 (74.0%)	
(+)	23 (20.2%)	5 (29.4%)	20 (26.0%)	
Molecular subtype				0.110
Luminal A	43 (37.7%)	7 (41.2%)	33 (42.9%)	
Luminal B	39 (34.2%)	8 (47.1%)	27 (35.1%)	
HER2	12 (10.5%)	2 (11.8%)	11 (14.3%)	
Triple-negative	20 (17.5%)	0 (0%)	6 (7.8%)	
Ki-67				0.266
Low	47 (41.2%)	8 (47.1%)	38 (49.4%)	
High	67 (58.8%)	9 (52.9%)	39 (50.6%)	
MR finding				0.002
Mass	99 (86.8%)	10 (58.8%)	55 (71.4%)	
NME	14 (12.3%)	6 (35.3%)	22 (28.6%)	
NA	1 (0.9%)	1 (5.9%)		
MR BPE				0.10
Minimal	76 (66.7%)	12 (70.6%)	36 (46.8%)	
Mild	26 (22.8%)	4 (23.5%)	31 (40.3%)	
Moderate	6 (5.3%)	0 (0%)	7 (9.1%)	
Marked	6 (5.3%)	1 (5.9%)	3 (3.9%)	
MR BPE				0.788
Minimal + Mild	102 (89.5%)	16 (94.1%)	67 (87.0%)	
Moderate + Marked	12 (10.5%)	1 (5.9%)	10 (13.0%)	

NA: not available. *: accurate vs. inaccurate

**Table 4 jcm-11-01172-t004:** Analysis of accurate and inaccurate groups for size measurement by radiologist.

Variables	Accurate 152 (73.1%)	Inaccurate 56 (26.9%)		*p*-Value *
		Underestimation 11 (5.3%)	Overestimation 45 (21.6%)	
Mean age (years)	58.01 ± 12.39	55.73 ± 12.39	56.76 ± 12.08	0.451
Age group				0.916
<50 years	45 (29.6%)	3 (27.3%)	14 (31.1%)	
≥50 years	107 (70.4%)	8 (72.7%)	31 (68.9%)	
Mean radiologist size (cm)	2.20 ± 1.42	2.50 ± 1.94	3.71 ± 1.92	<0.001
Mean pathologic size (cm)	2.11 ± 1.41	3.95 ± 2.65	2.52 ± 1.73	0.022
Pathologic size				0.065
0–2 cm	87 (57.2%)	1 (9.1%)	23 (51.1%)	
>2 cm	65 (42.8%)	10 (90.9%)	22 (48.9%)	
In situ component				0.017
(−)	50 (32.9%)	2 (18.2%)	7 (15.6%)	
(+)	102 (67.1%)	9 (81.8%)	38 (84.4%)	
EIC				0.008
(−)	109 (71.7%)	7 (63.6%)	18 (40.0%)	
(+)	21 (13.8%)	3 (27.3%)	11 (24.4%)	
NA	22 (14.5%)	1 (9.1%)	16 (35.6%)	
Histologic grade				0.358
Low	119 (78.3%)	9 (81.8%)	31 (68.9%)	
High	32 (21.0%)	2 (18.2%)	13 (28.9%)	
NA	1 (0.7%)		1 (2.2%)	
ER				0.279
(−)	35 (23.0%)	1 (9.1%)	16 (35.6%)	
(+)	117 (77.0%)	10 (90.9%)	29 (64.4%)	
PR				0.702
(−)	58 (38.2%)	3 (27.3%)	20 (44.4%)	
(+)	94 (61.8%)	8 (72.7%)	25 (55.6%)	
HER2				0.003
(−)	125 (82.2%)	8 (72.7%)	27 (60.0%)	
(+)	27 (17.8%)	3 (27.3%)	18 (40.0%)	
Molecular subtype				0.078
Luminal A	63 (41.4%)	5 (45.5%)	15 (33.3%)	
Luminal B	55 (36.2%)	5 (45.5%)	14 (31.1%)	
HER2	13 (8.6%)	1 (9.1%)	11 (24.4%)	
Triple-negative	21 (13.8%)	0 (0%)	5 (11.1%)	
Ki-67				0.537
Low	66 (43.4%)	6 (54.5%)	21 (46.7%)	
High	86 (56.6%)	5 (45.5%)	24 (53.3%)	
MR finding				<0.001
Mass	132 (86.8%)	7 (63.6%)	25 (55.6%)	
NME	19 (12.5%)	3 (27.3%)	20 (44.4%)	
NA	1 (0.7%)	1 (9.1%)		
MR BPE				0.519
Minimal	92 (60.5%)	7 (63.6%)	25 (55.6%)	
Mild	41 (27.0%)	3 (27.3%)	17 (37.8%)	
Moderate	11 (7.2%)	0 (0%)	2 (4.4%)	
Marked	8 (5.3%)	1 (9.1%)	1 (2.2%)	
MR BPE				0.274
Minimal + Mild	133 (87.5%)	10 (90.9%)	42 (93.3%)	
Moderate + Marked	19 (12.5%)	1 (9.1%)	3 (6.7%)	

*: accurate vs. inaccurate.

## Data Availability

Data are available on request from the corresponding author.

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
