# Peer review of "Evaluation of Breast Cancer Size Measurement by Computer-Aided Diagnosis (CAD) and a Radiologist on Breast MRI"

_jcm, 2022, doi:10.3390/jcm11051172_

Round 1

Reviewer 1 Report

The author summarizes and evaluates work on the cancer size measurement by CAD and radiologist by MRI. The author concludes that the “Radiologist-measured size was significantly more accurate than CAD size.”  Some comments are below:

It will be better to explain the abbreviations initially and avoid them in the abstract like EIC.

Do the authors think CAD measurement followed by a radiologist will be the best for measurement? For example, what will be the advantage if pathologist measurement and radiologist measurements are given weights to reduce the extent for “false-negative.”

For table 4, how does the subtype HER2 effect size determination

Is there any role of breast density in determination?

Author Response

Authors’ response to the reviewers’ comments

Reviewer 1

English language and style

(x) English language and style are fine/minor spell check required.

Comments and Suggestions for Authors

The author summarizes and evaluates work on the cancer size measurement by CAD and radiologist by MRI. The author concludes that the “Radiologist-measured size was significantly more accurate than CAD size.”  

Response: We would like to thank the reviewer for your constructive critique to improve the manuscript. We have made every effort to address the issues raised and to respond to all comments. Please, find next a detailed, point-by-point response to the reviewer's comments. We hope that our revisions will meet the reviewer’s expectations.

Some comments are below:

It will be better to explain the abbreviations initially and avoid them in the abstract like EIC.

Response: We would like to thank the reviewer for the question. Please note that we have explained the abbreviations initially, as per the reviewer’s suggestion.

Do the authors think CAD measurement followed by a radiologist will be the best for measurement? For example, what will be the advantage if pathologist measurement and radiologist measurements are given weights to reduce the extent for “false-negative.”

Response: We would like to thank the reviewer for the question. This study compared only CAD measurement and radiologist-measurement. I agree with you. It is very interesting to consider the CAD as a first reading followed by radiologist. The research on the comparison between CAD and combined measurement or radiologist and combined measurement will be required in the future. However, I think that combined measurement can be more likely to have advantages.

We have revised the limitation part in the manuscript as follows:

“Third, this study compared only CAD- and radiologist-measured size. In practice, combined measurement such as CAD as a first reading followed by radiologist can be helpful, so further study will be required in the future.” (Lines 393-395)

For table 4, how does the subtype HER2 effect size determination

Response: We would like to thank the reviewer for the question. We found that there was a mistake in the numerical notation in HER2 of the table 4, so it was corrected. The size measured by radiologist was more accurate for cancers with negative HER2 than positive HER2, as shown in lines 291-293.

Is there any role of breast density in determination?

Response: We would like to thank the reviewer for the question. Breast density may be an influencing factor, but we did not take into account the breast density of mammography. I think it can be included in future research.

Reviewer 2 Report

This study aimed to evaluate the size measurement of 208 breast cancers by a computer-aided diagnostic (CAD) system and a single radiologist on breast MRI regarding their histopathologic size to determine clinicopathologic and MRI factors that can affect the measurements, obtaining 11 size discrepancies in exact and inexact groups and concluding that the radiologist's measurement was more precise than the CAD. The submitted manuscript is a simple, interesting and well-presented study that, in my opinion, is of great interest in medical radiodiagnosis.
Minor considerations:
 Some additional aspects that do not decrease the interest of the study could be included in the discussion, such as:
- To describe the importance of the radiologist's experience in breast diagnosis by MRI and its incidence in this study carried out through an individual reading by a single radiologist.
- comment on the number of clinically non-palpable small tumors included in the study. Was there always a mass to be measured? Would it be 13.2% of the tumors in which the MR does not find a nodule/mass?
- incorporate the interest of the CAD as a first reading that can be reviewed later by the radiologist

Author Response

Authors’ response to the reviewers’ comments

Reviewer 2

English language and style

(x) I don’t feel qualified to judge about the English language and style.

Comments and Suggestions for Authors

This study aimed to evaluate the size measurement of 208 breast cancers by a computer-aided diagnostic (CAD) system and a single radiologist on breast MRI regarding their histopathologic size to determine clinicopathologic and MRI factors that can affect the measurements, obtaining 11 size discrepancies in exact and inexact groups and concluding that the radiologist's measurement was more precise than the CAD. The submitted manuscript is a simple, interesting and well-presented study that, in my opinion, is of great interest in medical radiodiagnosis.

Response: We would like to thank the reviewer for your constructive critique to improve the manuscript. We have made every effort to address the issues raised and to respond to all comments. Please, find next a detailed, point-by-point response to the reviewer's comments. We hope that our revisions will meet the reviewer’s expectations.

Minor considerations:
 Some additional aspects that do not decrease the interest of the study could be included in the discussion, such as:
- To describe the importance of the radiologist's experience in breast diagnosis by MRI and its incidence in this study carried out through an individual reading by a single radiologist.

Response: We would like to thank for your valuable comment. In this study, only one radiologist with 10 years of experience in breast imaging (mammography, ultrasound and MRI) measured all cancer size on breast MRI. This can be a limitation as well as advantage, avoiding interobserver variability. I think that further study including the comparison of cancer size measurement according to the breast radiologist’s experience will be required.

We have revised the limitation part in the manuscript as follows:

“Second, only one radiologist measured the cancer size on breast MRI. However, the radiologist had 10 years of experience in breast imaging and this can be an advantage, avoiding the interobserver variability.” (Lines 392-393)

- comment on the number of clinically non-palpable small tumors included in the study. Was there always a mass to be measured? Would it be 13.2% of the tumors in which the MR does not find a nodule/mass?

Response: We would like to thank the reviewer. In “3. Result” section, the range of pathologic size of cancers is 0.1-11.0 cm. Lesion types of breast cancer on MR were classified as mass or non-mass enhancement (NME) according to Breast Imaging Reporting and Data System. When no lesion (mass neither non-mass enhancement) was present on MRI or CAD, the size was set to 0 cm. Two cases was not present on MRI or CAD, which was marked as NA (not available) in MR finding part at Table 3 and 4.

- incorporate the interest of the CAD as a first reading that can be reviewed later by the radiologist

Response: We would like to thank the reviewer for the question. This study compared only CAD measurement and radiologist-measurement. I agree with you. It is very interesting to consider the CAD as a first reading followed by radiologist. The research on the comparison between CAD and combined measurement or radiologist and combined measurement will be required in the future. However, I think that combined measurement is more likely to have advantages.

We have revised the limitation part in the manuscript as follows:

“Third, this study compared only CAD- and radiologist-measured size. In practice, combined measurement such as CAD as a first reading followed by radiologist can be helpful, so further study will be required in the future.” (Lines 393-395)